# Fabrication of Carbon Nanomaterials Using Laser Scribing on Copper Nanoparticles-Embedded Polyacrylonitrile Films and Their Application in a Gas Sensor

**DOI:** 10.3390/polym13091423

**Published:** 2021-04-28

**Authors:** Yong-il Ko, Min-Jae Kim, Dong-Yun Lee, Jungtae Nam, A-Rang Jang, Jeong-O Lee, Keun-Soo Kim

**Affiliations:** 1Department of Physics and Graphene Research Institute, Sejong University, Seoul 05006, Korea; armist130@gmail.com (Y.-i.K.); mjkimphys@gmail.com (M.-J.K.); geovi012@gmail.com (D.-Y.L.); goodnjt@kist.re.kr (J.N.); 2Department of Electrical Engineering, Semyung University, Jecheon 27136, Korea; 3Advanced Materials Division, Korea Research Institute of Chemical Technology (KRICT), Gajeong-ro 141, Daejeon 34114, Korea; jolee@krict.re.kr

**Keywords:** carbon nanomaterials, laser scribing, gas sensor, polyacrylonitrile, stabilization, copper particles

## Abstract

Carbon nanomaterials have attracted significant research attention as core materials in various industrial sectors owing to their excellent physicochemical properties. However, because the preparation of carbon materials is generally accompanied by high-temperature heat treatment, it has disadvantages in terms of cost and process. In this study, highly sensitive carbon nanomaterials were synthesized using a local laser scribing method from a copper-embedded polyacrylonitrile (CuPAN) composite film with a short processing time and low cost. The spin-coated CuPAN was converted into a carbonization precursor through stabilization and then patterned into a carbon nanomaterial of the desired shape using a pulsed laser. In particular, the stabilization process was essential in laser-induced carbonization, and the addition of copper promoted this effect as a catalyst. The synthesized material had a porous 3D structure that was easy to detect gas, and the resistance responses were detected as −2.41 and +0.97% by exposure to NO_2_ and NH_3_, respectively. In addition, the fabricated gas sensor consists of carbon materials and quartz with excellent thermal stability; therefore, it is expected to operate as a gas sensor even in extreme environments.

## 1. Introduction

Carbon nanomaterials have excellent electrical, mechanical, thermal, and chemical properties, and they are applied in various fields, such as aerospace, construction, and sporting goods [1,2,3]. In particular, owing to the excellent molecular adsorption behavior resulting from conductivity and high specific surface area, their applications in the gas sensor field are promising [4,5,6]. However, in general, the synthesis of carbon nanomaterials has a disadvantage in terms of process time and cost, as it uses a high-temperature heat treatment at ~1000 °C or higher [3,7,8,9,10]. By contrast, the laser scribing method has the advantage of saving time and energy by enabling local carbonization in the desired pattern in a short period [11,12,13,14]. Polyacrylonitrile (PAN) is a well-known precursor of carbon materials and forms an intermediate in which nitrile groups are cyclized through stabilization pretreatment [15,16,17]. Since the laser scribing method can carbonize only cyclic polymers because of the fast process time and temperature changes [11], carbon nanomaterials can be synthesized from stabilized PAN via laser scribing, resulting in the desired shape and position.

In this study, PAN polymer, which is difficult to carbonize by laser scribing [11], could be successfully graphitized through stabilization pretreatment, and it is expected that by using this method, the type of polymer capable of laser scribing can be expanded. After coating the appropriate polymer above, direct pattern growth of carbon nanomaterials using laser scribing is possible, and it is expected that the application field can be expanded. Through this, as in this study, a directly pattered carbonization using laser scribing after coating with suitable polymer on various substrates will be possible, and it is expected that the application field can be expanded. For this purpose, the spin-coated thin PAN film was molecularly cyclized via stabilization, and carbon nanomaterials were patterned in the desired position and shape. This method was conducted as a process aimed to save time and minimize energy consumption compared to heat treatment using a self-produced LASER-assisted vacuum chamber. In addition, high-quality carbon nanomaterials were obtained by complexing with copper particles, which act as catalysts for carbonization [18,19,20]. The chemical and morphological properties of the synthesized material were evaluated using Raman spectroscopy, X-ray photoelectron spectroscopy (XPS) analysis, and scanning electron microscopy (SEM) image analysis, and the dynamic response of hazardous gases, such as nitrogen dioxide (NO_2_) and ammonia (NH_3_), was evaluated using a gas sensing chamber. The results confirmed that the pattern-synthesized carbon nanomaterial has a stereoscopic 3D structure that facilitates adsorption/desorption of gas molecules and, thus, exhibits excellent performance as a NO_2_ gas sensor. At this time, target gas was detected as changes in electrical properties due to the interaction with the carboxyl and hydroxyl groups of the carbon material [4,21,22,23,24,25], the synthesized material has such suitable disorder and oxygen group, which is advantageous for sensing. In particular, since the mixed copper provides free electrons, the sensor showed enhanced responsivity in NO_2_ detection than NH_3_, which operates by an electron donating effect of the sensor devise [26]. In addition, the prepared gas sensor consists of thermally stable carbon materials and quartz [12,27,28]; thus, it is expected to operate as a gas sensor even in extreme environments [29,30].

## 2. Experimental

### 2.1. Materials

PAN (Mw 150,000) and copper powder (<100 nm) were purchased from Sigma-Aldrich, Saint Louis, MO, USA. Dimethylformamide (DMF, 99.8%), ammonia water (25–30%), and quartz plates (15 mm × 15 mm × 1 mm) were purchased from SAMCHUN Chemical, DUKSAN Chemical, and HANJIN quartz, respectively.

### 2.2. Stabilization of Copper-Embedded PAN Thin Film

A schematic of the overall fabrication process is shown in Figure 1. First, PAN powder was added to the DMF solvent (PAN/DMF = 10 wt%) and stirred for 12 h. Cu/DMF solution was prepared by adding copper particles to DMF (10 wt%) and sonicated for 30 min. Further, the Cu and PAN solutions were mixed at a ratio of 0, 5, and 10 wt%, respectively. The prepared Cu-embedded PAN (CuPAN) solution was dropped onto a cleaned quartz substrate and spin-coated at 1000–1500 rpm for 30 s. The stabilized CuPAN (S-CuPAN) was prepared via evaporation and stabilization at 100 and 280 °C for 1 h, respectively (heating rate = 1 °C/min).

### 2.3. Laser Scribing on S-CuPAN and Lift-Off Untreated Area

A pulsed ytterbium fiber laser (MFP-20, MAXPHOTONICS Co., LTD., Shenzhen, China) was used to prepare the laser-scribed S-CuPAN (LS-CuPAN) under vacuum atmosphere (Appendix A). The specific parameters of the laser scriber are listed in Table 1. To prepare the optimized LS-CuPAN, scribing conditions such as laser power and repeat count were controlled. The distance of the laser spot was calculated using the scan rate and frequency, as shown in Equation (1). In this study, we fixed the frequency, scan rate, and spot spacing at 30 kHz, 450 mm/s, and 15 μm, respectively.
Distance (μm) = scan rate (mm/s)/frequency (kHz)(1)

The LS-CuPAN was immersed in ammonia water for 30 min to remove the unexposed area, and a locally patterned carbon nanomaterial channel (0.5 mm × 5 mm) was retained.

### 2.4. Gas Sensing

The LS-CuPAN gas sensor was measured to evaluate the dynamic response toward NO_2_ and NH_3_ gases using a source meter (Keithley 236, Cleveland, OH, USA) and switch system (Keithley 708A) (Appendix A). A 0.1 V DC potential was applied to the sensor, and changes were continuously monitored. The LS-CuPAN-based chemical sensor experiments were performed with chosen concentrations of the target gases diluted with air at a total gas flow of 500 sccm. In addition, the LS-CuPAN-based chemical sensor was measured at room temperature and atmospheric pressure. The injection and concentration of gases were automatically controlled by a mass flow controller, and the duration of the target and air as background gases were set to 10 and 50 min, respectively. The temperature of the chamber was maintained at 25 °C during the measurements. The normalized resistance response is defined as follows:ΔR/R_0_ (%) = (R − R_0_)/R_0_ × 100%(2)
where R_0_ and R are the resistance of the sensor in air and the target gas, respectively.

### 2.5. Characterization

Raman analysis and optical microscopy (OM) were performed using a Renishaw inVia Raman spectroscope with a 514 nm excitation line. The morphology and chemical analysis of the LS-CuPAN were characterized by a scanning electron microscope (SEM, VEGA3, TESCAN, Brno-Kohoutovice, Czech Republic) and an X-ray photoelectron spectroscope (XPS, K-alpha (Al Kα), Thermo Fisher, Waltham, MA, USA).

## 3. Results and Discussion

### 3.1. Effect of Stabilization of CuPAN

Figure 2a shows the Vis-NIR spectra and photographs of a CuPAN spin-coated on a quartz substrate at 1000 rpm before and after stabilization. The absorbance of CuPAN and S-CuPAN at a laser wavelength of 1064 nm were 0.13 and 0.77, respectively, and the inset photograph shows color changes from transparent brown before to black after stabilization. In addition, the appearance after laser scribing each film shows that laser-scribed CuPAN (L-CuPAN; without stabilization) exhibited almost non-carbonized parts, whereas LS-CuPAN was carbonized to a clear black color. This phenomenon occurred because the laser-exposed local region reached the carbonization energy after stabilization by significantly increased laser absorption. Furthermore, the embedded Cu nanoparticles were confirmed to act as the catalyst for carbonization.

As shown in Figure 2b, the Raman signal was not observed in CuPAN, which remained almost identical even after laser scribing (L-CuPAN). Lin et al. reported that laser-induced carbonization is not applicable to step- and chain-growth polymers but only to polymers with aromatic and imide repeat units because the laser scribing process is fast and accompanied by rapid changes in temperature [11]. Therefore, it was confirmed that pristine CuPAN with a linear chain did not carbonize by laser scribing. By contrast, since the PAN polymer can be partially carbonized and cyclized through stabilization pretreatment, it can be converted into a more easily carbonized form [15,16,17]. Therefore, in S-CuPAN, broad carbon-related peaks, such as first-order scattered G- and D-bands and second-order scattered 2D-band, were observed owing to the partial graphitic structure. Furthermore, exceptional G, D, and 2D-bands are visible in LS-CuPAN because complete carbonization by laser scribing from the stabilized form occurred. The improvement in Raman bands corresponding to the graphitic carbon structure owing to the combination of stabilization and laser scribing was verified.

### 3.2. Laser Scribing Conditions of LS-CuPAN

#### 3.2.1. Spin-Coating Conditions of CuPAN Thin Film

During spin-coating with the CuPAN solution (10 wt%), the rotational speed was changed from 1000 to 2000 rpm, and the resulting Vis-NIR spectra and thicknesses of the sample are shown in Figure 3a and Table 2, respectively (laser power = 4 W; repeat count = 1 time). The thicknesses at 2000, 1500, and 1000 rpm were 12, 19, and 23 μm, respectively, and the absorbance increased to 0.18, 0.37, and 0.77, respectively (at 1064 nm). Since this absorbance changed the laser absorption of the precursor, it directly affected the local carbonization quality during laser scribing.

Figure 3b and Table 2 show the Raman spectra, calculated I_D_/I_G_, I_2D_/I_G_, and full width at half maximum (FWHM) of the G + D′-band after laser scribing. First, I_D_/I_G_ was 1.43, and the 2D-band was almost not observed at 2000 rpm; thus, it was confirmed as an amorphous graphitic structure containing considerable disorder. This phenomenon is a result of a decrease in laser absorption owing to the low thickness. By contrast, the Raman spectra of LS-CuPAN at 1000 rpm exhibited distinctive G-, D-, and 2D-band, and I_D_/I_G_ decreased to 0.94, which is lower than that of I_D_/I_G_ (1.02) at 1500 rpm. In addition, the FWHM of the G + D′-band was calculated to be 64.2 cm^−1^, which is less than 1500 and 2000 rpm, and this value was caused by a decrease in the disorder-induced D′-band, located on the right shoulder of the G-band, and an increase in the uniform carbon bonding. Therefore, since LS-CuPAN at 1000 rpm has a high laser absorption, an enhanced graphitic structure was verified to form with less disorder [31,32,33].

#### 3.2.2. Weight Ratio of Cu Particle to PAN

Raman spectra of S-CuPAN and LS-CuPAN when the weight ratio of Cu to PAN is 0, 5, and 10 wt% are shown in Figure 4a and Appendix A (rotational speed = 1000 rpm, laser power = 4 W, repeat count = 1 time). As shown in the spectrum of S-CuPAN (10 wt%), carbon-related bands are observed because they are partially carbonized through stabilization even without laser scribing, as mentioned above. However, because of a high I_D_/I_G_ (1.45) and a weak 2D-band (I_2D_/I_G_ = 0.26), it is considered an amorphous structure containing numerous disorders. It has similar characteristics at both 0 and 5 wt%. By contrast, after laser scribing, I_D_/I_G_ gradually decreased to 1.13, 1.08, and 0.94, and I_2D_/I_G_ increased to 0.16, 0.33, and 0.51 with weight ratios of 0, 5, and 10 wt%, respectively. In addition, since the FWHM of the G + D′-band also decreased from 100.1 to 64.2 cm^−1^, the disorder decreased and the uniformity of carbon bonding increased. Therefore, high-quality carbon was formed at the higher ratio of Cu/PAN, and it was confirmed that Cu particles acted as a catalyst through the absorption of laser energy during the carbonization of PAN [18,19,20]. However, at a high Cu ratio, the viscosity increased rapidly, and spin-coating was not performed well. Thus, 10 wt% was considered the most appropriate concentration. Additional chemical analysis of the concentration of Cu particles will be discussed later in the XPS analysis.

#### 3.2.3. Optimization of the Laser Scribing Process (Power, Repeat Time)

Figure 4b and Table 2 show the Raman spectra and calculated I_D_/I_G_, I_2D_/I_G_, and FWHM of the G + D′-band according to different laser powers with LS-CuPAN (10 wt%) (rotational speed = 1000 rpm; repeat count = 1 time). As the laser power increased from 2 to 4 W, the sharpness of carbon-related scattering such as G, D, and 2D bands increased and was observed. Furthermore, I_D_/I_G_ decreased from 1.37 to 0.94 due to a decrease in D-band, and I_2D_/I_G_ increased from 0.11 to 0.51 due to an increase in 2D-band. However, when the laser power was increased above 4 W, the carbon-related bands broadened. In the case of 6 W, I_D_/I_G_ was increased to 1.75, and the 2D-band almost disappeared. These results indicate that the graphitic structure was decomposed, and an amorphous structure was formed because of excessive laser exposure. Moreover, the FWHM of the G + D′-band also had the lowest value of 64.2 cm^−1^ with 4 W, but the FWHM increased as the laser power decreased or increased based on 4 W. Consequently, the quality of carbon increased as the laser power increased, and LS-CuPAN with the least disordered graphitic structure was formed at 4 W. Decomposition occurred above that value because of excessive energy accumulation.

Figure 5 shows photographs, optical images (OM), and Raman spectra according to the repeat count of the laser scribing. First, as shown in the photograph of Figure 5a, S-CuPAN was carbonized to the clearest black when the scribing count was 1, and this effect decreased as the repeat counts increased three and five times. In addition, considering this as OM, the scribing part is carbonized black in Figure 5b. However, when the repeat count increased three (Figure 5c) and five times (Figure 5d), the carbonized part disappeared; consequently, the quartz substrate was exposed five times. These results can also be described using the Raman spectra in Figure 5e. As the repeat count increased from 1 to 5, the I_D_/I_G_ increased from 0.94 to 1.64, and the 2D band almost disappeared. Therefore, since an increase in repeat count causes decomposition by inducing excessive energy accumulation, scribing only once is considered an optimal condition. Consequently, the laser scribing conditions for synthesizing the carbon nanomaterial pattern with the best quality were optimized for 4 W and one time.

### 3.3. Chemical and Morphological Analysis of LS-CuPAN

#### 3.3.1. X-ray Photoelectron Spectroscopy (XPS)

To confirm the chemical composition of the surface of the synthesized material, XPS measurements were performed, as shown in Figure 6. To confirm the carbonization effect during laser scribing, the C1s spectra were peak fitted. The narrow-scan C1s spectra of S-CuPAN (Figure 6a) show that the main peak is 285.5 eV with a broad shoulder at higher binding energy. The deconvoluted C1s spectra show that they are composed of functional groups of C=C(sp^2^), C–C(sp^3^), C–N, C–O, and C=O located at 285.2, 285.7, 286.9, 287.8, and 288.7 eV, respectively. In particular, the C–N bond peak derived from the nitrile group of PAN was observed, and the specific portion of the sp^3^ C–C, C–O and C=O bonds was relatively large. By contrast, the C–N peak almost disappeared in the spectra of LS-CuPAN (Figure 6b), and the relative area of the sp^2^ C=C bond increased because of carbonization, indicating a maximum intensity at 285.1 eV. Furthermore, FWHM was significantly reduced from 3.3 to 1.5, and the distribution of the chemical bonds of carbon is more uniform than that of S-CuPAN. These results confirmed that LS-CuPAN was synthesized into a graphitic carbon structure via laser scribing.

Figure 6c shows the atomic ratio (at %) of C1s according to the weight ratio (wt%) of Cu to PAN. When the ratio of Cu was 5 and 10 wt%, the atomic ratio of C1s increased by 13.62 and 19.70% after laser scribing, respectively. This implies that the higher the proportion of Cu, the higher the carbonization efficiency. Cu is a well-known catalytic transition metal for carbonization [18,19,20]. Therefore, the carbonization efficiency was confirmed to be improved by increasing the Cu ratio owing to the catalytic effect, thereby increasing the atomic ratio of carbon. Appendix A shows the Cu2p spectra of S-CuPAN (a) and LS-CuPAN (b). As shown in this figure, peaks corresponding to Cu2p_1/2_ and Cu2p_3/2_ were observed in S-CuPAN, and these peaks were observed even after laser scribing in LS-CuPAN. These results indicate that the Cu particles were not decomposed and acted as catalysts during laser scribing.

#### 3.3.2. Scanning Electron Microscopy (SEM)

The macromorphology of the prepared LS-CuPAN was observed via SEM, and the images are shown in Figure 7. The low-magnification image (Figure 7a) shows LS-CuPAN with a diameter of 500 μs and quartz substrate after the removal of untreated S-CuPAN. Figure 7b,c show SEM images with high magnification and tilting by 70°. As illustrated by these images, LS-CuPAN is a stereoscopic 3D structure formed as an aggregate of carbon nanomaterials. Since this porous 3D structure contains numerous interspaces and voids, the accessible surface area of the material can be increased [34,35,36]. Therefore, it is considered suitable as a gas sensor material because it can improve the adsorption/desorption of gas molecules.

### 3.4. Gas Sensing Property

NO_2_ is an air pollutant emitted from combustion during industrial and recycling processes, and NH_3_ also has harmful effects on the human body, causing skin and respiratory diseases. Therefore, it is critical to detect gaseous substances to prevent damage from these hazardous gases.

Figure 8 shows the dynamic response of LS-CuPAN after exposure to 10, 50, and 100 ppm of NO_2_ and NH_3_ gases at room temperature (25 °C) and pressure (760 Torr). In general, since the oxidizing NO_2_ molecule accepts electrons from oxygen on the sensor surface, the hole conductivity of the sensor increases. By contrast, in reducing NH_3_ molecules, the lone-pair electrons of nitrogen donate electrons and the conductivity of the sensor decreases [36]. Consequently, during exposure to 10, 50, and 100 ppm of NO_2_ (NH_3_) gas environments for 10 min, the resistance responses were detected as −1.03 (0), −1.71 (0.76), and −2.41 (0.97)%, respectively (Appendix A. Resistance changing rates exposure to different gas concentrations and Appendix A. Resistance response versus time). This resistance response occurred as a result of the effective carbonization of CuPAN due to improved absorbance and cyclization by stabilization as seen in Vis-NIR, and synthesized LS-CuPAN has a carbon crystal structure having a suitable oxygen-derived disorder that induces detection of NO_2_ and NH_3_, as shown in XPS and Raman results. Meanwhile, since the operation of gas sensors is closely related to the adsorption/desorption behavior of gas molecules on the surface, a three-dimensional (3D) structure is more advantageous than a flat 2D structure [33,34,35]. Therefore, this sensing performance is caused by the porous 3D structure of LS-CuPAN, as mentioned above. In addition, when Cu nanoparticles are complexed with carbon material, more free electrons are supplied to improve the chemical adsorption of NO_2_ [37], confirming that the resistance response of NO_2_ is higher than that of NH_3_.

## 4. Conclusions

In summary, we successfully prepared a highly sensitive LS-CuPAN gas sensor from the carbonization of a CuPAN composite via laser scribing, following spin-coating. The CuPAN thin film fabricated on the quartz substrate was converted into a carbonization precursor through stabilization and then patterned into a carbon nanomaterial using a pulsed laser with the desired shape and position. In particular, the stabilization process was essential in laser-induced carbonization because of enhanced absorbance and cyclization, and the addition of copper promoted this effect as a catalyst. In this study, the optimal conditions for rotational speed for spin-coating, weight ratio of Cu, laser power, and repeat count were 1000 rpm, 10 wt%, 4 W, and one time, respectively. The synthesized LS-CuPAN showed that I_D_/I_G_, I_2D_/I_G_, and FWHM of G + D′ were 0.94, 0.51, and 64.2 in Raman results, respectively, and 79.2 at% of C1s was included through XPS. Moreover, it had a stereoscopic 3D structure that was easy for gas sensing, and when exposed to 10, 50, and 100 ppm of NO_2_ (NH_3_) gas environments for 10 min, the resistance responses were detected as −1.03 (0), −1.71 (0.76), and −2.41 (0.97)%, respectively. Therefore, the LS-CuPAN-based gas sensor was more sensitive in NO_2_ than NH_3_ molecules.

In short, this study provided an effective method for manufacturing a carbon material that can be patterned into the desired shape through a simple process. In addition, since the fabricated LS-CuPAN consists of quartz and carbon material with excellent thermal stability, it is expected to operate as a gas sensor even in extreme environments.

## Figures and Tables

**Figure 1 polymers-13-01423-f001:**
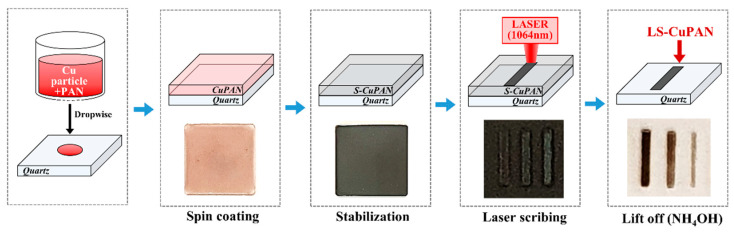
Schematic of the synthesis process and photographs of LS-CuPAN.

**Figure 2 polymers-13-01423-f002:**
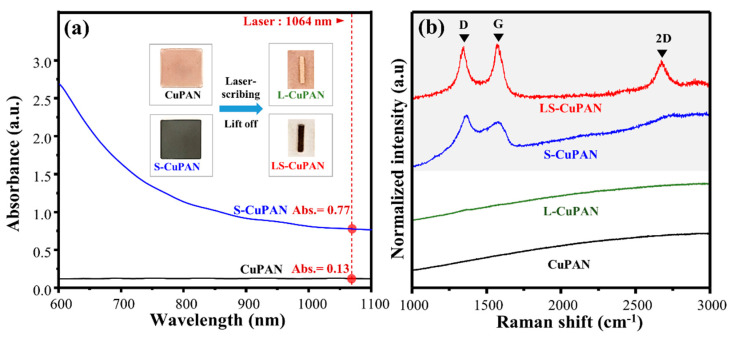
(**a**) Vis-NIR spectra and photographs of CuPAN and S-CuPAN and (**b**) Raman spectra of CuPAN, L-CuPAN, S-CuPAN, and LS-CuPAN.

**Figure 3 polymers-13-01423-f003:**
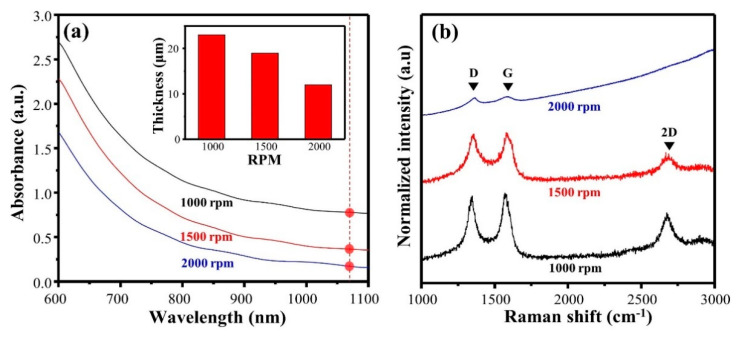
(**a**) Vis-NIR spectra and thickness of S-CuPAN and (**b**) Raman spectra of LS-CuPAN depend on spin-coating conditions.

**Figure 4 polymers-13-01423-f004:**
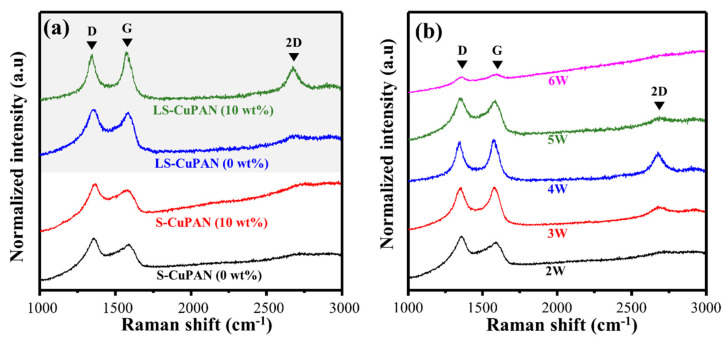
Raman spectra with different (**a**) weight ratios of Cu/PAN and (**b**) laser power.

**Figure 5 polymers-13-01423-f005:**
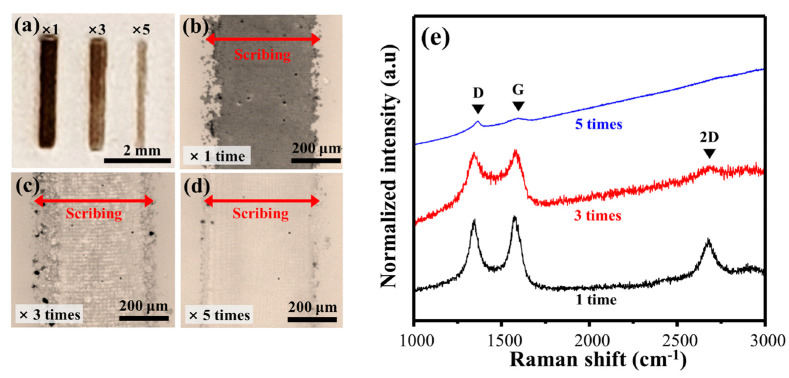
(**a**) Photograph, (**b**–**d**) optical images, and (**e**) Raman spectra of LS-CuPAN with different exposure repeat times.

**Figure 6 polymers-13-01423-f006:**
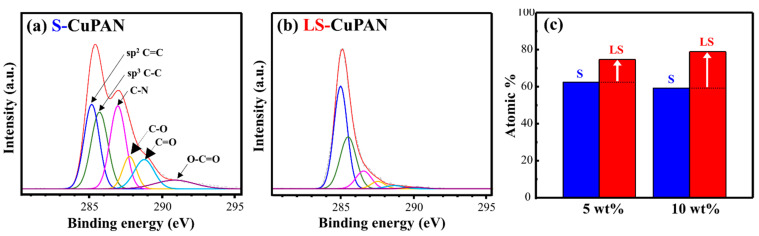
C1s spectra of (**a**) S-CuPAN (10 wt%), (**b**) LS-CuPAN (10 wt%), and (**c**) atomic ratio of carbon in XPS analysis.

**Figure 7 polymers-13-01423-f007:**
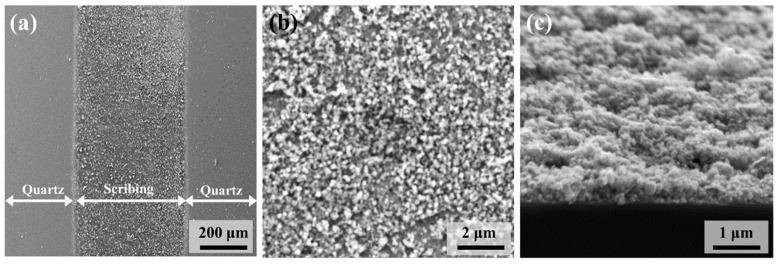
SEM images of LS-CuPAN in (**a**,**b**) top and (**c**) tilted (70°) view.

**Figure 8 polymers-13-01423-f008:**
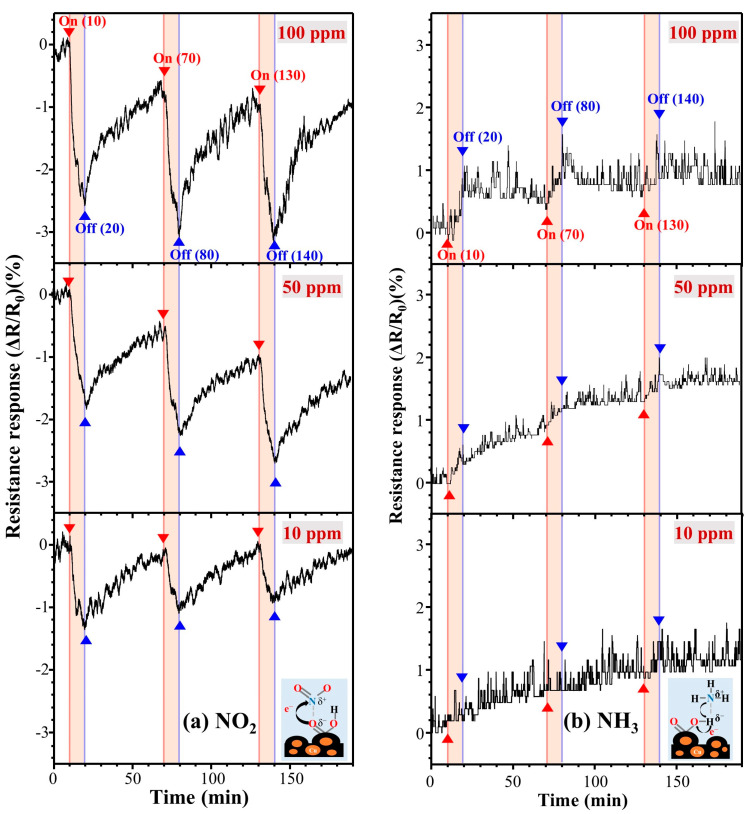
Resistance response curves exposure to different concentrations. (**a**) Under NO_2_ 10, 50, and 100 ppm, (**b**) under NH_3_ 10, 50, and 100 ppm.

**Table 1 polymers-13-01423-t001:** Specific parameters of laser scriber (MFP-20).

Parameter	Wavelength (nm)	Laser Power (W)	Scan Speed (mm/s)	Beam Diameter (μm)	Frequency (kHz)
Data	1064 ± 4	1–20	0–1500	6–9	30–60

**Table 2 polymers-13-01423-t002:** Specific synthetic conditions of the LS-CuPAN and analytic results calculated from Raman spectroscopy.

Stabilization	Cu/PAN Ratio (wt%)	Spin Coating (rpm)	Thickness (μm)	Absorbance (a.u.)	Laser Power (W)	Repeat Count (times)	I_D_/I_G_	I_2D_/I_G_	FWHM of G + D′ (cm^−1^)
X	10	1000	31	0.13	4	-	n/a	n/a	n/a
O	10	1000	23	0.77	4	-	1.45	0.26	111.5
O	0	1000	20	0.60	4	1	1.13	0.16	100.1
O	5	1000	22	0.74	4	1	1.08	0.33	82.6
O	10	1000	23	0.77	4	1	0.94	0.51	64.2
O	10	1000	23	0.77	4	1	0.94	0.51	64.2
O	10	1500	19	0.37	4	1	1.02	0.43	79.2
O	10	2000	12	0.18	4	1	1.43	n/a	107.7
O	10	1000	23	0.77	2	1	1.37	n/a	114.5
O	10	1000	23	0.77	3	1	1.02	0.25	87.3
O	10	1000	23	0.77	4	1	0.94	0.51	64.2
O	10	1000	23	0.77	5	1	1.09	0.23	94.8
O	10	1000	23	0.77	6	1	1.75	n/a	109.3
O	10	1000	23	0.77	4	1	0.94	0.51	64.2
O	10	1000	23	0.77	4	3	1.08	0.25	92.7
O	10	1000	23	0.77	4	5	1.64	n/a	118.3

## Data Availability

Data will be made available at request.

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
