# Peer review of "Fabrication of Carbon Nanomaterials Using Laser Scribing on Copper Nanoparticles-Embedded Polyacrylonitrile Films and Their Application in a Gas Sensor"

_polymers, 2021, doi:10.3390/polym13091423_

Round 1

Reviewer 1 Report

The author represented a highly sensitive carbon nanomaterial for a gas sensor using a local laser scribing method from CuPAN composite film. This paper has a lot of discrepancies regarding gas sensing measurement. The author should address those comments before acceptance in Polymers journal, my comments are as follows.

 1.Introduction should be modified by including few recent reports on NH3 and NO2 sensing and few advantages of using CuPAN as NH3 and NO2 sensors.

2. NH3 and NO2 gases are not VOCs.

2. Author should explain the mixing of the gases using MFCs with a schematic of the sensing chamber.

2. What is the baseline value of the sensors in terms of current. Also, the noise is excessively high in Fig.8. Also, include a concentration dependence sensing measurement. 

3. The sensitivity is too low compare to other sensors. Also, response and recovery times are high.

4. Author should address the structural effect on sensing using XPS, Raman, and UV-VIS spectra. 

5. Conclusion should be modified.

Reviewer 2 Report

Dear authors,

your paper intitled "Fabrication of carbon nanomaterials using laser scribing on copper nanoparticles-embedded polyacrylonitrile films and their application in a gas sensor" is interesting in the fabrication of the sensor (laser-based), but there are some doubts about the current content devoted to the characterizations and to the experimental stages approach under NO2 and NH3 gases.

Major revisions:
- the measurement chain, the gas chamber, the flowmeters, gas reservoirs, are missing. There is no the description of the system able to characterize the feasibility of the sensors;
- Figure 8 and the general content of the paper: for readers, not so expert in this chemical sensors area, couldn't appreciate the +0.5% of current response under 100 ppm NO2. 100 ppm of NO2 is a strong amount of gas (very dangerous if dispersed in the environment for the human health). The authors should add the gain of the responses versus time. Please start by adding values of time in the figure 8 (pink column - from on to off). Then, add a new table or discussion about the rise time: very important!!!;
- there is no trace about the responses of the sensors under 1-5-10 ppm of NO2 (in the literature 100 ppm of NO2 is never considered due to very high concentration), especially for the application of the gas per 1-5 minute(s). To my side, the response to NH3 is negligible due to the negative drift by time. You can improve the characterization under NH3 if you increase the time of gas application;
- table 2: you proposed some important factors for the fabrication of the sensors. The levels of factors didn't follow a statical-methodological approach (i.e. Design of Experiments). It seems a screening. Please underline in the paper why you're proposing a screening approach, that is the early stage for a project. 

Minor revisions:
- at the end of the "Introduction" section: add a summary of the following sections;
- figure 5: add the unit of measure and therefore the clearance of each scribing width;
- figure 8b: check for the y label "reponse".

Best regards.

Take care.

Round 2

Reviewer 1 Report

The author made significant changes in the manuscript. In the present form, this manuscript can be accepted for publication in Polymers Journal.  

Reviewer 2 Report

Dear author,

the paper is now accepted.

Take care.